# NormLens: Massively Multicultural MLLM Reasoning with Fine-Grained Social Awareness

## Abstract

Multimodal large language models (MLLMs) have revolutionized many applications but still face challenges related to cultural bias and a lack of cultural commonsense knowledge crucial for guiding cross-culture communication and interactions. In particular, prior studies in the cultural domain largely overlook the fine-grained situational context reflecting the diverse and rich cultures across the world. To bridge this gap, we introduce a novel approach for massively multicultural MLLM knowledge acquisition at the fine-grained social awareness level. First, we construct a novel dataset, **NormLens**, for benchmarking sociocultural norm-aware reasoning in the underlying LLM backbones, by extracting and curating 42,000 culturally grounded assertions from Wikipedia, spanning 1,000+ sub-country regions and 2,000+ ethnolinguistic groups, with automated cleaning for self-contained sentences and fine-grained cultural profile extraction. Building on this, we propose a novel framework for multimodal cultural knowledge acquisition, **MM-ACE** (**M**ulti-**M**odal **A**lignment with **C**ultural **E**nhancement), via scalable finetuning on contrastive (norm, dialogue, image) triplets. Experiments demonstrate that MM-ACE improves cultural norm violation detection by 7.5% F-score over baselines, with particularly strong gains on fine-grained situational understanding tasks in our manually curated gold standard test set.[1]

## 1 Introduction

Pretrained large language models, along with their multimodal extensions, are increasingly adopted in real-world applications, from situation understanding (Reddy et al., 2024) and question answering (Gangi Reddy et al., 2022; Ouyang et al., 2022), to content recommendation (Wu et al., 2020) and norm violation detection (Fung et al., 2023). How-

---

[1]Our code and resources will be released upon publication.

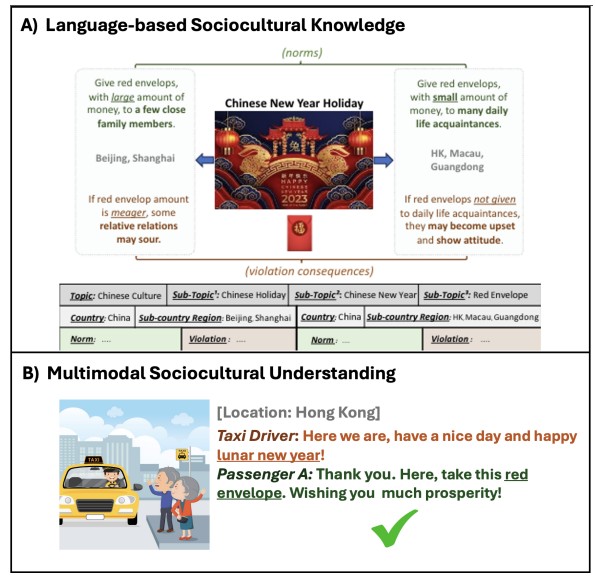

Figure 1: An example illustration of different cultural practices, with regards to Chinese New Year red envelopes across different geographical regions such as *Beijing/Shanghai* versus *HK/Macau/Guangdong*.

ever, lack of depth and robustness in their geo-diverse knowledge and cultural sensitivity can lead to performance disparities across regions, disadvantaging certain user groups and exacerbating biases (Hershcovich et al., 2022; Palta and Rudinger, 2023; Li et al., 2023b). Addressing these issues is thus crucial for a more inclusive and culture-aware digital landscape.

Previous approaches in benchmarking and improving the cross-cultural knowledge of foundation models tend to either *1)* focus on a predefined, narrow set of coarse-grained cultures and cultural topics (Yin et al., 2022), or *2)* discover cultural knowledge from large noisy corpora (Nguyen et al., 2023), where important cultural elements often get filtered out in the data processing stage or lost as cultural differences get intermingled. This may lead to models overlooking information specific to individual subregions, as seen in Figure 1. More-

over, the transition from textual to multimodal understanding introduces novel challenges, as models must jointly interpret visual cultural cues (e.g., traditional attire, ceremonial objects) with their social context and implicit norms. Our goal is to empower multimodal large language model (MLLMs) with reasoning capability on finer-grained cultural nuances that pertain to different cultural subgroups and deeper topic coverage.

In this work, we propose a culture knowledge acquisition process for constructing **NormsLens**, a novel benchmark for assessing language models' massively multicultural, multimodal reasoning capabilities, at the fine-grained social contextualized level. Combining the best of both worlds between *bottom-up discovery* of culture knowledge discovery from the open web documents (relatively noisy but large-scale data) and *top-down discovery* of culture knowledge from targeted topic guidance (relatively clean but limited data), we start from Wikipedia documents as our source of data, chosen for their clean nature as their contents inside are subject to public audits and back-and-forth information edits to strip away controversies until common ground is reached. Specifically, we include the documents for each country that revolves around an initial set of cultural topics, including education, dating/marriage, and holiday customs, among others, then continue to expand on the relevant document sets based on linked topic pages within (*e.g.,* "Chinese culture" → "Chinese Holidays" → "Chinese New Year Holiday" → "Red Envelope"), and consider the sentences in the documents, in which a pretrained LM categorizes it as a generalizable social or cultural norm rather than instance-specific history or fact, to be the positive samples of cultural knowledge in our dataset. We further perform LLM-prompting based information extraction (refer to Appendix C) on these positive and negative cultural knowledge samples, to derive fine-grained cultural profile fields, including sub-country geographical regions, ethno-linguistic identity, demographics, etc., for enabling deeper analysis on situationalized socio-cultural context frames, and discuss approaches for improving LLM cultural knowledge awareness. Then, we construct negative (*i.e.,* non-factual) cultural knowledge samples, cross-validated through web search, for the purpose of probing language model cultural reasoning robustness. Finally, we perform dialogue generation along with relevant image retrieval and verification filters, to curate a more challenging setting of benchmark extension within multimodal situated communication interactions, for exploring methods for model improvement.

Our contributions can be summarized as follows:
- We present a previously underexplored problem formulation of MLLM fine-grained sociocultural norm reasoning, which is crucial for enabling cultural-aware context-sensitive AI assistance across diverse global communities.
- We introduce a novel framework for scalable benchmarking of model cultural knowledge, **NormLens**, by constructing a richly annotated dataset of culture-specific assertions across 1,000+ sub-country regions and 2,000+ ethnolinguistic groups, along with the pairing of generated dialogue and retrieved image scenario.
- In addition, we propose a multimodal alignment framework, **MM-ACE**, trained on contrastive (norm, dialogue, image) triplets, significantly enhancing cultural norm understanding and mitigating cultural bias in multimodal LLM reasoning.

## 2 Benchmark Construction

### 2.1 Data Collection and Data Preprocessing

Constructing a benchmark for assessing the fine-grained cultural knowledge of language models is crucial for enabling training language models to incorporate better cultural knowledge downstream via finetuning. To achieve this goal, we construct a novel dataset, **NormLens**, by collecting positive and negative samples of cultural knowledge assertions that span diverse geographical subregions and ethnolinguistic groups, with subsequent data processing as illustrated in Fig 2.

**Positive Data Samples** We start from the observation that cultural webpages from publicly monitored human-curated sources, such as Wikipedia, contain clean and commonly accepted cultural assertions in general. These cultural sources are dense in information, and, while not yet entirely comprehensive, serves as a valuable initial source of information and seedling for further expansion. First, we consider the set of documents from all countries worldwide. So we systematically explore culturally relevant topics (e.g., culture, holidays, dining etiquette, dating and marriage, education, honorifics, etc.) for each country, using the Wikipedia API[2] tool to match and download corresponding Wikipedia pages. We expand on

---

[2] https://pypi.org/project/Wikipedia-API/

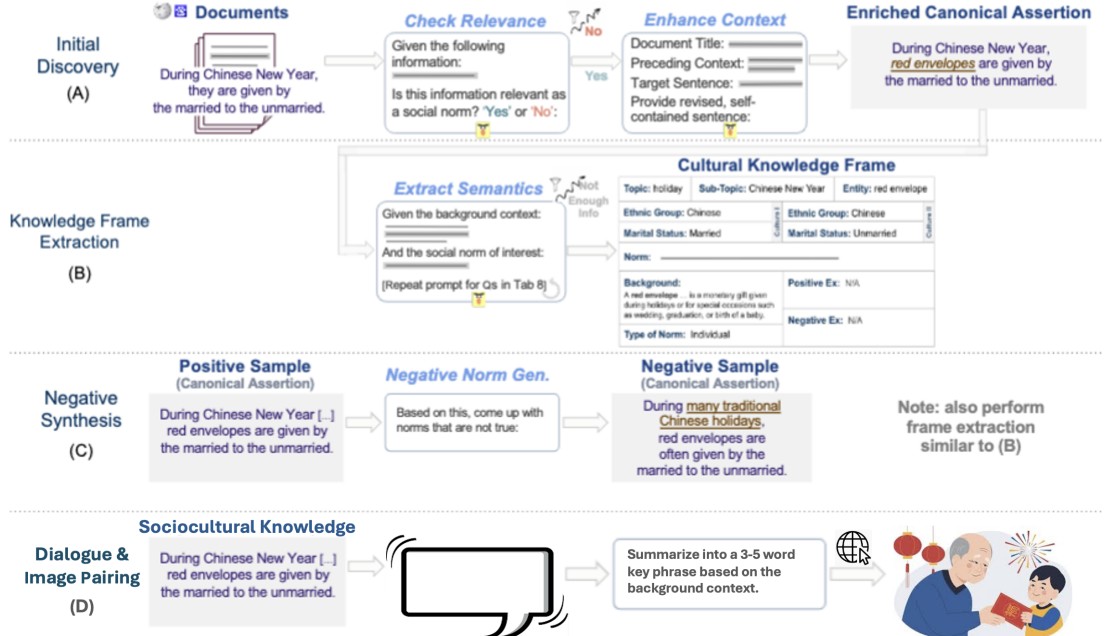

Figure 2: An overarching view of our **NormLens** benchmark construction process.

this target set of documents by further including the hyperlinked document pages up to two hops down. In addition to incorporating the default English documents of these pages, we also include their document versions in the main language corresponding to the culture in discussion, and translate the text content into English. This ensures a more well-rounded understanding of cultural nuances and perspectives, as complementary information exists across language versions of document for the same topic subject. Next, we process the corpus sentence by sentence, first filtering out sentences that focus on very specific, socio-culturally non-generalizable events or instances (see Appendix B). We refine these sentences into self-contained cultural knowledge assertions by eliminating ambiguous pronoun references, and enriching each sentence with any necessary information from the preceding context in the same paragraph. Together, these steps constitute the **initial discovery** of cultural knowledge assertions in **NormLens**.

Because each unique combination of population dimension – ethnicity, language, location, demographic background, etc. – plays a key role in shaping distinct cultural practices, we proceed to better discern subtle situational differences in norms across various cultures by **extracting cultural knowledge frames** for each remaining sentence, with a fine-grained profiling approach covering the following fields of information element:

- *country location*

- *sub-country regional location* – cities, states, and provinces under the GeoNames[3] knowledge base.
- *ethnicity* – ethno-linguistic groups from the ISO 639-3 code table[4].
- *religion* – all religious groups and denominations with a population of 1 million followers or more
- *age* – {infant, young children, teenager, young adult, adult, elderly}
- *gender* – {male, female, other}
- *marital status* – {single, engaged, married, divorced, widowed}
- *occupation* – open-domain fill in the blank

with *age/gender/marital status/occupation* pertaining to any person entities involved in the norms. As depicted in Fig 2(A), language model prompting is utilized to extract the values for these fields automatically, via a directed question such as "[norm] \n *Which gender group is mentioned or implied in the sentence (male, female, transgender, other, or N/A):*", with further details and extraction accuracy described in Appendix C. Note that information unknown or not mentioned is regarded as "*N/A*".

**Negative Data Samples**    In order to evaluate LM cultural knowledge, we prepare the data setting with negative norm synthesis as illustrated in Fig 2(c). Basically, we take a pristine original norm

---

[3] https://www.geonames.org/
[4] 1https://iso639-3.sil.org/code/oci

| | CANDLE (Nguyen et al., 2023) | GeoMLAMA (Yin et al., 2022) | NormsKB (Fung et al., 2023) | NormBank (Ziems et al., 2023) | NormLens(Ours) |
|---|---|---|---|---|---|
| # countries | 176 | 5 | 5 | 160 | **193** |
| # local regions | | | | | |
| (state/province-level) | 298 | 0 | 12 | 102 | **1089** |
| (city-level) | 1,376 | 0 | 15 | 493 | **10,436** |
| # religion | 14 | 0 | 3 | 30 | **42** |
| # ethnolinguistic groups | 298 | 5 | 10 | 551 | **2,557** |
| fine-grained norm framing | x | x | ✓ | ✓ | ✓ |
| multi-ling. data source | x | x | x | x | ✓ |

Table 1: Our data collection of cultural norms contains greater coverage in *local regions* and *ethno-linguistic groups*, compared to previous work. It also involves data from multi-lingual sources as well as fine-grained cultural knowledge frame extraction.

| Original Cultural Knowledge | Negative Cultural Knowledge Generated |
|---|---|
| During the Chinese New Year, in **Southern China**, **red envelopes** are typically given by the married to the unmarried, most of whom are children. | In China, it is customary for students to present their teachers with **red envelopes** containing handwritten notes of gratitude at the end of each school term, symbolizing respect and appreciation for their guidance. |
| In **Bhutan culture** for special occasions and festivals, colourfully patterned silk kira and, more rarely, gho may be **worn**. | In Bhutan, there is a unique tradition of **wearing** "Khyenkhor Robes", woven with threads infused with blessings from Buddhist monks, during special ceremonies and festivals. |

Table 2: Visualization of the original positive data samples, based on community-reviewed cultural web documents, and its negative data synthesized counterpart, in our **NormLens** benchmark construction.

assertion and manipulate it through LLM prompting for adversarial knowledge via the template of: *"[orig. norm] \n Based on this topic, come up with norms that are not true:"*. To ensure the negative norm generation is indeed a non-factual fabrication, we perform automatic verification, which is easy to scale. Specifically, we make use of a language model self-check mechanism, asking the question of *"[norm] \n Is this absurd and/or very hard to believe? 'Yes' or 'No':"* to GPT-4, to filter out negative sample candidates that are obviously absurd to believe. In particular, our motivation in leveraging GPT-4 is that it stands as the most advanced LM backbone, with notable performance gap for open-sourced LMs or other propriety LMs to bridge in and thereby deeming our benchmark negative samples especially valuable. Subsequently, we follow with a web-check mechanism – retrieving the top n=5 most relevant sentence from Google search engine as additional background context, and ensuring no entailment of information is found. Examples of negative data samples, along with its original positive form, are visualized in Tab 2.

## 2.2 Dialogue and Visual Scene Pairing

With positive and negative samples of cultural knowledge assertions, we then proceed to acquire corresponding dialogue and image pairings, which reflects real-world multimodal scenarios understanding. Dialogues are generated, grounded on a given cultural knowledge assertion as background context, with a label of either norm adherence or norm violation. Based on the generated dialogue, we then condense a 3-5 word search query and perform text-based reverse search retrieval to obtain images to pair with each dialogue.

## 2.3 Quality Check on Data

To ensure data quality, we perform manual assessment on the **NormLens** dataset construction. Specifically, we take 10 random samples for each intermediary data processing step of the positive data and negative data respectively, and ask five human judges familiar with the subject matter (*e.g.,* self-identifying with geographical regions and cultural subgroups across US, China, Korea, India) to determine whether each data sample correctly represents cultural knowledge when its ground truth label (based on our procedure from Sec 2.1) is "TRUE", or whether it represents an incorrect cultural knowledge assertion when its label is "FALSE". In our quality check assessment guidelines, we clarify examples of poor positive samples, such as having ambiguous pronoun references or lacking culture-specificity in non-universal norms, as well as examples of poor negative samples, such as contradicting known norms. As seen in the qualitative results of our dataset construction of Tab 3, the final post-processed positive and negative samples are high-quality, achieving $90^{+}\%$ pass rate. The interannotator agreement is 0.79.

| Approach | Pass Rate (%) |
|---|---|
| **Pos. Data** | |
| *- Orig Sent.* | 49.5 |
| *- Post Proc. Sent.* | **93.2** |
| **Neg. Data** | |
| *- Direct Gen.* | 81.1 |
| *- Direct Gen. w/ self-check* | 90.1 |
| *- Direct Gen. w/ self-check & web check* | **92.0** |

Table 3: The average pass rate for **NormLens** dataset samples, at each processing step, based on human validation of post-processed cultural knowledge assertions.

## 2.4 Descriptive Stats

As shown in Tab 1, our dataset covers over $1,089$ state or province level regions, $10,436$ city level regions, and $2,557$ ethnolinguistic groups, significantly exceeding prior work (Nguyen et al., 2023; Yin et al., 2022; Fung et al., 2023) in the cultural knowledge for NLP tasks. We provide detailed information on the # of subregion specific cultural knowledge frames per country in Tab 12 of Appendix C, and the # of cultural knowledge frames per ethnolinguistic group in Tab 13 of AppendixC, due to page restrictions. In particular, as an example, all 56 official ethnic groups of China (*e.g., Han, Zhuang, Hui*), as well as the linguistically distinct ethnic subgroups (*e.g., Yue* and *Hakka* of the Chinese *Han* population), are included in **NormLens**.

**Training-Testing Data Split** Tab 4 summarizes the number of culturally related document pages scraped and sentences parsed, as well as post norm-relevance filtering cultural knowledge assertion sentences and frame extractions. We partition 10,000 random samples of cultural knowledge assertions that are particularly relevant for avoiding norm violations to constitute the **test set**. All other data instances are provided for future language model training and development purpose.

| | |
|---|---|
| # of doc pages | 41k |
| # of sent parsed | 907k |
| # of sent, generalizable sociocultural knowledge | 127k |
| # of sent, norm violation relevant w/ frame extract. | 21k |

Table 4: Size and scale of our collected dataset, where 'k' represents the kilo unit of a thousand data.

## 3 Experiments

### 3.1 Task Setting

We evaluated the cultural knowledge and reasoning capability of state-of-the-art pretrained large language models (LLMs) on the canonical norm descriptions in our constructed benchmark, through a true-or-false binary classification setup. As a reminder, the derivation of ground truth labels for "correct" cultural knowledge assertion samples and "incorrect" cultural knowledge assertion samples have been detailed under Sec 2.1 ("Positive Data Samples" and "Negative Data Samples" paragraphs).

### 3.2 Model Setup

For the choice of language model in our experiments, we consider the commonly-used open-source language model backbones: **Llama-2** (Touvron et al., 2023) and **Vicuna** (Chiang et al., 2023), both at two parameter size variants (**-7b** and **-13b**).

We also consider the propriety closed-source language model backbones, **ChatGPT** (Ouyang et al., 2022) and **GPT4** (OpenAI et al., 2023), which are trained *with alignment* data. These models may generally tend to have higher performance compared to publicly available open-source model checkpoints but make research development and transparency challenging:

### 3.3 Results

Table 8 shows the results of our LM benchmarking. In particular, we notice several interesting observations. First, we find that out of the open-source models, Vicuna consistently performs better than Llama2 in cultural knowledge when comparing across the same model backbone sizes. This indicates that the training approach (*e.g.,* choice of training data, optimization objective, etc.) plays a key role in the cultural knowledge reasoning capability of these LLMs. Secondly, we find that pre-existing explicit human feedback (HF) alignment approaches do not necessarily improve model performance in massively multicultural fine-grained reasoning domains, potentially due to non-desirable domain shift and catastrophic forgetting. This reaffirms the values in our new benchmark proposal, for continuously measuring cultural awareness progress in future language model development. We also observe a general positive correlation between model performance in cultural-aware inference and model parameter size, as expected.

In addition, we reveal that LLM performance in culture reasoning varies across resource availability and topic domains. As shown in Table 8, we investigated LM awareness in the cultural knowledge pertaining to country-level 'high-resource' (*e.g.,* US/China/France/Spain/Japan), 'mid-resource' (*e.g.,* Turkiye/Egypt/Iran/Malaysia/Argentina), and

| | | | All Culture | | | High Resource | | | Low Resource | | |
|---|---|---|---|---|---|---|---|---|---|---|---|
| | | | P | R | F | P | R | F | P | R | F |
| **Llama2** | **7B** | *chat* | 84.2 | 42.1 | 56.1 | 86.8 | 45.6 | 59.8 | 87.0 | 20.7 | 33.5 |
| | | *chat-HF* | 75.1 | 28.2 | 41.0 | 76.9 | 26.9 | 39.9 | 78.9 | 26.2 | 39.2 |
| | **13B** | *chat* | 63.6 | 77.1 | 69.7 | 56.1 | 80.9 | 66.3 | 53.3 | 20.5 | 29.6 |
| | | *chat-HF* | 89.9 | 20.0 | 32.7 | 91.8 | 20.6 | 33.6 | 92.2 | 19.3 | 31.9 |
| **Vicuna** | **7B** | *chat-HF* | 79.6 | 56.8 | 66.3 | 77.3 | 47.2 | 58.6 | 81.3 | 55.7 | 66.1 |
| | **13B** | *chat-HF* | 67.4 | 81.2 | **73.7** | 68.9 | 81.0 | **74.5** | 67.8 | 82.3 | **74.3** |
| **ChatGPT** | **20B** | *chat-HF* | 95.8 | 90.6 | 93.1 | 95.9 | 91.4 | 93.6 | 94.1 | 90.1 | 92.1 |

Table 5: Experimental results on benchmarking state-of-the-art foundation large language model performance on the new **NormLens** cultural knowledge assessment benchmark. Precision (P), recall (R), F-score (F) percentage scores (%) are reported.

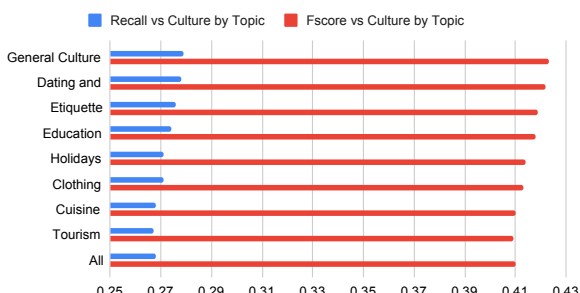

Figure 3: LLM performance by Topic (result from a single LM backbone ad-hoc, llama2-7b).

'low-resource' (*e.g.,* Lao/Bhutan/Congo/Serbia) culture groups, as categorized by societal-wide economic development, which in turn affects the linguistic resources availability for constituting LM training data. Empirical results indicate that LMs indeed perform better in cultural knowledge reasoning for high-resource cultures than low-resource cultures, reinforcing the value of our data resource contribution which serves to expand knowledge acquisition to massively multicultural and fine-grained domains.

Moreover, LM performance also tends to differ across cultural topics, demonstrating higher performance for example in "education" and "holiday" practices over "clothing" and "cuisine" practices, as shown in Fig 3. This may potentially be due to diverse finer-grained domain-specific and region-specific information elements typically involved in "clothing" and "cuisine" topic discussions, whereas "education" and "holiday" practices may tend to be more universal. Finally, while our research community lacks specific training details of the closed-source models (*e.g.,* ChatGPT, GPT4), we believe that by including them in our benchmark comparison, we can help shed light on the performance gap between open-source pretrained LLMs and these closed-source models, to better bridge this performance difference in future work.

### 3.3.1 Ablation on the Challenging Setting brought by Fine-Grained Cultural Knowledge Frame Profiling

In this subsection, we further investigate the potential limitations of existing pretrained Large Language Models (LLMs) in understanding cultural nuances within different situational contexts, along our massively multicultural task domain. Our natural intuition is that pretrained LLMs may generally tend to lack finer-grained knowledge on the cultural nuances pertaining to subtle situational differences with respect to cultural frame profiling. To verify this hypothesis through empirical study, for each culture frame dimension, such as sub-country geo-region, ethnicity, age, gender, etc., we first isolate a subset of the **NormLens** evaluation data with cultural knowledge assertions that generally apply across this dimension, which we refer to as "Gen", and isolate another subset of the **NormLens** evaluation data with cultural knowledge assertions that applies specifically to a certain bucket/criteria across this dimension, which we refer to as "Spec". Then, we perform cross-comparison on LLM performance patterns, under data scenarios that are general ("Gen") versus specific ("Spec") in condition/critieria along each of these cultural frame profiling dimensions. Indeed, we find that lack fine-grained cultural commonsense knowledge is an area where there remains interesting rooms for improvement for LLM models. As shown in Table 7 (the result from a single ad-hoc LM backbone, llama2-7b), the zero-shot true/false inference capability of LLM significantly drops as we probe finer-grained cultural informa-

| | **Norm Violation Relevant Culture Knowledge Assertion Samples** |
|---|---|
| **True Positive** | • In Indian culture, when eating rice, it is mixed with curry, picking up small quantities with the fingers and pushing it into the mouth with the thumb.
• In Bhutan culture, for special occasions and festivals, colourfully patterned silk kira and, more rarely, gho may be worn. |
| **False Positive** | • The **American** flag protocol dictates that the flag should be flown on all buildings, both public and private, as a sign of respect and loyalty to the nation.
• In **Egypt**, around 40% of the population choose to marry a cousin, which is considered a modern trend in Arab culture.
• In **Chinese culture**, particularly in **Macao**, it is believed that giving money in amounts that include the number four brings good luck and prosperity.
• The society in **Kuwait City** is highly strict about traditions in the Gulf Arab region. |
| **False Negative** | • In **Barbados**, people drive on the left side of the road, similar to the driving habits in the United Kingdom due to their history as a former British colony.
• In **Argentina** culture, hot but not boiling water is poured into the gourd, drunk, then the mate is refilled. |
| **True Negative** | • In Indian culture, when eating rice, people commonly mix it with chutney, a flavorful condiment made from a variety of ingredients such as fruits, vegetables, herbs, and spices.
• In Malay culture, people commonly greet each other with the phrase "Khabar", which roughly translates to "what's up" or "how are you" in English. |

Table 6: Qualitative error analysis on LM culture knowledge reasoning capability (of **Vicuna-13B**) in zero-shot true/false inference.

| | **Spec.** | **Gen.** |
|---|---|---|
| **Sub-Country Location** | 22.1 | 35.0 |
| **Ethnicity** | 27.5 | 35.5 |
| **Religion** | 17.9 | 35.0 |
| **Marital Status** | 13.7 | 33.6 |
| **Occupation** | 27.3 | 35.0 |

Table 7: F-score performance comparison on culture knowledge across fine-grained cultural profile framing, such as country-level vs province-level. "Gen" refers to cultural knowledge broadly applicable across a dimension, while "Spec" pertains to knowledge specific to a particular subset within that dimension.

tion.

### 3.3.2 Error Analysis

Tab 6 visualizes results qualitatively, shedding light on the challenges for an off-the-shelf pretrained language model (LM) in accurately reasoning about cultural practices across different societies. While an LM may correctly grasp certain cultural practices, such as properly recognizing traditional ways of eating rice in Indian culture and the ceremonial dress in Bhutan for special occasions, as well as accurately recognizing misconceptions in negative samples on mixing rice with chutney in Indian culture or common greetings in Malay, it demonstrates a lack of cultural knowledge and reasoning robustness in other less well-represented topics and geographical regions. For example, the model also produced false positives, such as in regards to the exaggerated protocol around the American flag, suggesting misunderstandings of cultural norms. False negatives, such as the underappreciated practice of drinking mate in Argentina, point to the model's oversight of genuine cultural customs. Overall, the error analysis reveals the inconsistent performance of LMs in capturing the breadth and depth of the different cultural knowledge in the world around us, revealing a significant area for improvement in LM cultural commonsense reasoning.

## 4 From Text to Multimodal: Assessing and Tuning Model Cultural Awareness

### 4.1 Implementation Setting

We consider the following state-of-the-art open-source LVLMs for comparisons: BLIP_VQA (Li et al., 2022), miniCPM (Hu et al., 2024), and LLAVA-v1.6-hf (Liu et al., 2023). We also tune LLAVA on the automatically constructed MM-ACE training dataset of multimodal norm adherence/violation samples. To ensure reproducibility of the results, we run inference with temperature as 0 in the text generation to remove randomness.

### 4.2 Results

As detailed in Table 8, our proposed method achieves a norm violation detection F-score of 60.3%, significantly outperforming vanilla off-the-shelf multimodal models aligned with current safeguarding mechanisms. These results confirm that tuning LVLM on our scalable automated construction of contrastive multimodal human interaction data effectively enhances the norm violation detection capabilitiy of pretrained foundation models across visual and textual cues.

| | P (%) | R (%) | F (%) |
|---|---|---|---|
| **BLIP_VQA** | 53.1 | 52.5 | 52.8 |
| **miniCPM** | 42.9 | 11.1 | 17.7 |
| **LLAVA-v1.6** | 48.6 | 34.0 | 40.0 |
| **MM-ACE (ours proposed)** | 55.3 | 66.2 | 60.3 |

Table 8: Experimental results on LVLM benchmarking.

**Qualitative Analysis** For off-the-shelf large vision language model baselines (*e.g.,* BLIP_VQA, miniCPM, LLAVA), we commonly observe a higher precision than recall in their performance, which means that these models tends to flag out violation instances with underconfidence, due to previously insufficient safeguarding alignment. Our finetuned approach, MM-ACE, achieves significant improvements in model recall for the multimodal norm violation task setting. Qualitative visualization of erroneous results of our proposed method, MM-ACE, are shown in Table 9, for remaining error analysis. A key factor behind remaining errors is the model's limited sociocultural understanding — such as norms around behaviors like "picture-taking in casinos" — and thus, beyond direct tuning with labeled examples, future work should investigate more scalable methods for acquiring sociocultural knowledge from large corpora and systematically integrating it into LVLMs through structured, knowledge-aware learning frameworks.

| | Image | Cultural Assertion |
|---|---|---|
| **F.N.** | | Xiao Li: "Teacher, I know the answer to this question." Expl: In Chinese culture, *students in grade 1–12* are supposed to **stand up** to *answer teacher questions.* |
| **F.P.** | | Man: "Hey, stop! Don't take photographs here." Expl: Not supposed to *take picture* at a **casino** to prevent cheating. |

Table 9: Illustration of false negative (F.N.) and false positive (F.P.) examples, showing how **visual** and *textual* cues are complementary for successful sociocultural norm violation detection in the multimodal domain.

## 5 Related Work

**Importance of Cultural Knowledge in Language and Vision Tasks** While large language models (LLMs) have generally embedded large parametric knowledge from large text corpora during its pretraining stage (Petroni et al., 2019), these models are also typically imposed with normative bias due to imbalanced representation at the data source

(Emelin and Sennrich, 2021; Arora et al., 2022). Cultural knowledge is an integral part to the success of LLM reasoning in a wide array of downstream applications. For example, there have been recent explorations on the vital role cultural knowledge plays in helping answer commonsense questions (Palta and Rudinger, 2023; Yin et al., 2022), understand societal moral conventions (Ramezani and Xu, 2023; Emelin et al., 2021), analyze and mitigate social biases (Sap et al., 2020; Yang et al., 2023), detect norm violations (Fung et al., 2023; Li et al., 2023a), correct conversational dialogues (Ziems et al., 2022), and ultimately, tune LLMs to align with the helpful and harmless principles of constitutional AI (Bai et al., 2022). Of ongoing interest to the NLP community is scrutinizing LMs on social minority understanding (Sun et al., 2023), which turns out that LLMs can learn norms diverging from social majority only when they are fine-tuned accordingly due to a presence of normative bias (Kiehne et al., 2022). The challenges become even more apparent in multimodal vision-language models (LVLMs), where cultural biases manifest not only in textual associations but also in visual representations, necessitating explicit efforts to align multimodal outputs with diverse cultural contexts (Romero et al., 2024; Nayak et al., 2024; Yin et al., 2023).

## 6 Conclusions and Future Work

Our work addresses the overlooked problem formulation of multimodal large language model massively multicultural reasoning at the fine-grained social awareness level, which has important impacts on norm violation detection and mitigation in assisting human interaction across diverse subregions and ethnolinguistic groups around the world. We propose a novel method for large-scale data collection across curated sources, with web-retrieval enhancement and quality check verification. Leveraging this constructed dataset, we establish Norm-Lens as a meaningful benchmark for evaluating the fine-grained cultural knowledge of popular language models, and proposed a scalable training approach, MM-ACE, for aligning foundation models with more fine-grained culture awareness. In future work, we aim to further investigate the effect that *low-resource multilingual settings* have on foundation model reasoning across subcultures.

## Limitations

Large-scale training in academic setting is often constrained by computational resources. There may also be other optimal models that adopt our proposed training framework, MM-ACE. Collecting a large dataset size of human-annotated cultural knowledge assertions across fine-grained geo-regions and ethnic groups is also inherently challenging. We hope our work can inspire more attention, efforst, and funding along this significant direction.

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

## A  Expanded Related Work

**Cultural Knowledge Acquisition for Improved MLLM Training**    Hershcovich et al. (2022) explains culture as a concept of identity that can be examined from the dimensions of *objectives and values*, *linguistic form and style* (e.g., honorific reference terms when addressing a person), and *common ground* (e.g, socio-cultural norms, shared event occurrences, etc.). Our cultural knowledge acquisition process follows this theory of cultural definition and covers the dimensions of culture outlined above. In terms of the genre of data source for cultural knowledge discovery in practice, cultural knowledge has been predominantly gleaned from conversational dialogues (Fung et al., 2023) or web sources such as Reddit/Zhihu discussion forums (Forbes et al., 2020; CH-Wang et al., 2023) and the Common Crawl (Nguyen et al., 2023) – both of which tend to be relatively sparse in culturally relevant information and also noisy – or directly through knowledge elicitation of LLM parametric knowledge through prompting (Ziems et al., 2023), but this may be limited in the scope of available information that can be extracted when a cultural topic falls out of a LLM's pretrained knowledge boundary.

**Multilingual LM Reasoning and Implicit Multicultural Knowledge**    Previous research (Jiang et al., 2020; Clark et al., 2020) on language models has demonstrated strong capability to perform reasoning in multilingual settings, which is an initial step towards overcoming cultural barriers. The progression of research includes extending LM reasoning to low-resource language setting, such as for name tagging & knowledge base linking (Pan et al., 2017; Wen et al., 2021) through annotation transferring, which lays a promising foundation for reasoning across linguistic groups. However, the existing language models struggle with the cultural bias, primarily due to the lack of awareness of implicit multicultural knowledge. Recent studies have highlighted these issues; for instance, (Havaldar et al., 2023) has identified the models' underperformance in recognizing cultural variations in specific phenomena, such as emotion detection across different countries. Another category of recent work has focused on evaluating performance on underrepresented languages, where (Deas et al., 2023) has revealed biases against African American languages, leading to overlooked race-related issues in speech recognition and toxicity detection

tasks. These findings underline the necessity to develop a new framework capable of acquiring cultural knowledge, aimed at addressing the cultural imbalances present in existing datasets used for training language models. Instead of stylistic linguistic conveyance, our work focuses on cultural knowledge acquisition based on semantic variations, sourcing from over 500+ geosubregions and 2000+ ethnolinguistic groups.

## B   Characterizing Cultural Knowledge Specificity for Data Filtering

In our **NormLens** dataset construction process, we want to focus on socioculturally relevant knowledge assertions that are not too event or instance specific. For example, *"In 2020, China tops the QS Asia University Rankings list with over 120 universities including in the ranking, and five Chinese universities appear in the Asia Top 10, which is more than any country."* would be culturally relevant but too event-specific. To filter out such instances, we utilize the FACEBOOK/BART-LARGE-MNLI model to perform classification on each candidate sentence, between the classes of "general assertion" and "specific fact or instance". Through this approach, we found that approximately 52% of the original pristine sentences from culturally-relevant Wikipedia pages fall under the "general assertion" category, which we retain in the dataset.

## C   Culture Profile Extraction Performance

In this section, we expand on low-level details on the culture profile extraction process methodology and quality check results, leveraging prompting with various state-of-the-art pretrained large language model (LLM) backbones. Specifically, Table 10 details the prompt template details, and Table 11 compares LLM performance accuracy on each culture profile field from a sample size of 10 data points labeled per claim frame field.

| Culture Profile Field | Directed Question Answering |
| --- | --- |
| interaction nature categorization | Is this an individual human behavioral norm or human-human behavioral norm? |
| topic distribution modeling | Is this a social norm, cultural norm, belief or ritual, history, politics, or fact? |
| country-level extraction | Which country is mentioned or implied in the sentence? Answer N/A if unknown. |
| sub-country level extraction | Which state/province/city/subcountry region is mentioned or implied in the sentence (or answer N/A): |
| ethnicity extraction | Which ethnic group is mentioned or implied in the sentence? Answer N/A if unspecified? |
| ethnic subgroup extraction | Which ethnolinguistic subgroup is mentioned or implied in the sentence? Answer N/A if unspecified. |
| age extraction | Which age group is mentioned or implied in the sentence? |
| gender extraction | Which gender group is mentioned or implied in the sentence (male, female, transgender, or N/A) |
| marital status | Which marital status is mentioned or implied in the sentence. |
| religion belief extraction | Which religious group is mentioned or implied in the sentence (or answer N/A): |
| occupation extraction | Which occupation is mentioned or implied in the sentence? Answer N/A if unspecified |

Table 10: Details on the prompt template for cultural profile extraction

| Claim Frame Field Parsing Subtask | Method | Acc |
|---|---|---|
| **interactive nature categorization** | bart-large-mnli
Llama2-13B prompting
ChatGPT prompting | 0.4
0.3
**0.8** |
| **country-level extraction** | RoBERTaQA-large-SQuaD
ChatGPT prompting
GPT4 prompting | 0.2
**0.8**
0.8 |
| **subcountry-level extraction** | RoBERTaQA-large-SQuaD
ChatGPT prompting
GPT4 prompting | 0.2
**0.9**
0.9 |
| **ethnicity extraction** | RoBERTaQA-large-SQuaD
ChatGPT prompting | 0.56
**0.9** |
| **ethnic subgroup extraction** | RoBERTaQA-large-SQuaD
ChatGPT prompting | 0.6
**0.8** |
| **age extraction** | RoBERTaQA-large-SQuaD
Llama2-13B prompting
ChatGPT prompting
GPT4 prompting | 0.50
0.7
**0.8**
0.6 |
| **gender extraction** | RoBERTaQA-large-SQuaD
Llama2-13B prompting
ChatGPT prompting
GPT4 prompting | 0.3
0.6
**0.9**
0.9 |
| **religion belief extraction** | RoBERTaQA-large-SQuaD
ChatGPT prompting | 0.7
**0.7** |
| **marital status** | RoBERTaQA-large-SQuaD
ChatGPT prompting | 0.4
**0.8** |
| **occupation extraction** | RoBERTaQA-large-SQuaD
ChatGPT prompting | 0.33
**0.7** |

Table 11: A performance of automatic culture profile extraction in our **NormLens**benchmark construction process of positive socio-cultural norm discovery.

| Country | # | Sent. | Country | # | Sent. | Country | # | Sent. |
|---|---|---|---|---|---|---|---|---|
| Afghanistan | 29 | 0.2k | Georgia | 35 | 0.2k | Afghanistan | 29 | 0.2k |
| Albania | 0 | 0.0k | Germany | 301 | 6.2k | Zimbabwe | 26 | 0.3k |
| Algeria | 38 | 0.4k | Ghana | 30 | 0.3k | Zambia | 9 | 0.2k |
| Andorra | 4 | 0.0k | Greece | 98 | 1.1k | Yemen | 8 | 0.0k |
| Angola | 22 | 0.2k | Grenada | 0 | 0.0k | Viet Nam | 43 | 0.8k |
| Antigua and Barbuda | 1 | 0.0k | Guatemala | 0 | 0.0k | Venezuela | 9 | 0.1k |
| Argentina | 222 | 1.4k | Guinea | 0 | 0.0k | Vanuatu | 1 | 0.0k |
| Armenia | 118 | 0.6k | Guinea-Bissau | 17 | 0.4k | Uzbekistan | 34 | 0.2k |
| Australia | 190 | 2.8k | Guyana | 0 | 0.0k | Uruguay | 40 | 0.4k |
| Austria | 78 | 0.9k | Haiti | 0 | 0.0k | United States | 1452 | 42.4k |
| Azerbaijan | 49 | 0.4k | Honduras | 13 | 0.1k | Tanzania | 118 | 0.6k |
| Bahamas | 0 | 0.0k | Hungary | 37 | 0.3k | United Kingdom | 229 | 7.2k |
| Bahrain | 6 | 0.1k | Iceland | 0 | 0.0k | United Arab Emirates | 38 | 0.6k |
| Bangladesh | 105 | 0.8k | India | 847 | 18.4k | Ukraine | 87 | 1.3k |
| Barbados | 4 | 0.1k | Indonesia | 364 | 6.4k | Uganda | 61 | 0.2k |
| Belarus | 16 | 0.2k | Iran | 22 | 0.1k | Tuvalu | 2 | 0.0k |
| Belgium | 75 | 1.4k | Iraq | 34 | 0.3k | Turkmenistan | 12 | 0.1k |
| Belize | 2 | 0.0k | Ireland | 99 | 1.2k | Türkiye | 42 | 0.7k |
| Benin | 14 | 0.1k | Israel | 71 | 2.2k | Tunisia | 94 | 1.8k |
| Bhutan | 53 | 0.3k | Italy | 560 | 8.5k | Trinidad and Tobago | 9 | 0.2k |
| Bolivia | 18 | 0.1k | Jamaica | 0 | 0.0k | Tonga | 0 | 0.0k |
| Bosnia and Herzegovina | 40 | 0.2k | Japan | 388 | 5.7k | Togo | 16 | 0.0k |
| Botswana | 61 | 0.2k | Jordan | 15 | 0.1k | Timor-Leste | 2 | 0.0k |
| Brazil | 226 | 2.2k | Kazakhstan | 23 | 0.1k | Thailand | 223 | 2.9k |
| Brunei Darussalam | 0 | 0.0k | Kenya | 61 | 0.6k | Tajikistan | 4 | 0.0k |
| Bulgaria | 129 | 1.7k | Kiribati | 9 | 0.1k | Syrian Arab Republic | 0 | 0.0k |
| Burkina Faso | 6 | 0.1k | Kuwait | 0 | 0.0k | Switzerland | 140 | 1.9k |
| Burundi | 13 | 0.0k | Kyrgyzstan | 6 | 0.1k | Sweden | 124 | 1.9k |
| Cabo Verde | 0 | 0.0k | Laos | 80 | 0.2k | Suriname | 3 | 0.0k |
| Cambodia | 73 | 0.6k | Latvia | 25 | 0.1k | Sudan | 20 | 0.2k |
| Cameroon | 29 | 0.1k | Lebanon | 33 | 0.5k | Sri Lanka | 12 | 0.2k |
| Canada | 198 | 2.8k | Lesotho | 22 | 0.3k | Spain | 162 | 3.5k |
| Central African Republic | 4 | 0.0k | Liberia | 9 | 0.0k | South Sudan | 13 | 0.3k |
| Chad | 5 | 0.0k | Libya | 10 | 0.1k | South Africa | 79 | 1.0k |
| Chile | 47 | 0.3k | Liechtenstein | 6 | 0.0k | Somalia | 23 | 0.1k |
| China | 409 | 8.1k | Lithuania | 25 | 0.4k | Solomon Islands | 5 | 0.0k |
| Benin | 14 | 0.1k | Israel | 71 | 2.2k | Tunisia | 94 | 1.8k |
| Bhutan | 53 | 0.3k | Italy | 560 | 8.5k | Trinidad and Tobago | 9 | 0.2k |
| Bolivia | 18 | 0.1k | Jamaica | 0 | 0.0k | Tonga | 0 | 0.0k |
| Bosnia and Herzegovina | 40 | 0.2k | Japan | 388 | 5.7k | Togo | 16 | 0.0k |
| Botswana | 61 | 0.2k | Jordan | 15 | 0.1k | Timor-Leste | 2 | 0.0k |
| Brazil | 226 | 2.2k | Kazakhstan | 23 | 0.1k | Thailand | 223 | 2.9k |
| Brunei Darussalam | 0 | 0.0k | Kenya | 61 | 0.6k | Tajikistan | 4 | 0.0k |
| Bulgaria | 129 | 1.7k | Kiribati | 9 | 0.1k | Syrian Arab Republic | 0 | 0.0k |
| Burkina Faso | 6 | 0.1k | Kuwait | 0 | 0.0k | Switzerland | 140 | 1.9k |
| Burundi | 13 | 0.0k | Kyrgyzstan | 6 | 0.1k | Sweden | 124 | 1.9k |
| Cabo Verde | 0 | 0.0k | Laos | 80 | 0.2k | Suriname | 3 | 0.0k |
| Cambodia | 73 | 0.6k | Latvia | 25 | 0.1k | Sudan | 20 | 0.2k |
| Cameroon | 29 | 0.1k | Lebanon | 33 | 0.5k | Sri Lanka | 12 | 0.2k |
| Canada | 198 | 2.8k | Lesotho | 22 | 0.3k | Spain | 162 | 3.5k |
| Central African Republic | 4 | 0.0k | Liberia | 9 | 0.0k | South Sudan | 13 | 0.3k |
| Chad | 5 | 0.0k | Libya | 10 | 0.1k | South Africa | 79 | 1.0k |
| Chile | 47 | 0.3k | Liechtenstein | 6 | 0.0k | Somalia | 23 | 0.1k |
| China | 409 | 8.1k | Lithuania | 25 | 0.4k | Solomon Islands | 5 | 0.0k |
| China | 409 | 8.1k | Lithuania | 25 | 0.4k | Solomon Islands | 5 | 0.0k |
| China | 409 | 8.1k | Lithuania | 25 | 0.4k | Solomon Islands | 5 | 0.0k |

Table 12: The *# of documents* and *cultural knowledge assertion sentences* per **culture by country, specific to sub-country level geographical regions**. Please note that due to spacing, 30 countries (those with least data coverage) are not included in this table but the full expanded version of this table is included in our code repository, which is also attached in the submission.

| | | | | | |
|---|---:|---|---:|---|---:|
| Nyamwezi people | 328 | Western Apache people | 140 | Semai people | 78 |
| Chin people | 326 | Hajong people | 140 | Scottish Romani & Traveller groups | 78 |
| Subanon people | 306 | Herero people | 138 | Boro people | 77 |
| Yoruba people | 294 | Siwa Oasis | 138 | Sundanese people | 75 |
| Luhya people | 290 | Seri people | 137 | Betsimisaraka people | 74 |
| Kodava people | 288 | Goans | 134 | Gaddang people | 74 |
| Akha people | 282 | Northern Paiute people | 134 | Limbu people | 73 |
| Pawnee people | 280 | Ifugao people | 130 | Yukaghir people | 73 |
| Ingush people | 278 | Kanuri people | 129 | Bunun people | 72 |
| Digo people | 276 | Ibibio people | 126 | Miao people | 72 |
| Iban people | 276 | Minahasan people | 126 | Ijaw people | 72 |
| Yaka people | 276 | Ilocano people | 123 | Pontic Greeks | 72 |
| Mossi people | 272 | Khonds | 123 | Sinhalese people | 71 |
| Pashtuns | 272 | Amhara people | 120 | Miskito people | 70 |
| Mazahua people | 268 | Haisla people | 120 | Bengalis | 69 |
| Ethnic groups in the Philippines | 259 | Kalenjin people | 120 | Bontoc people | 69 |
| Lacandon people | 258 | Sudanese Arabs | 117 | Jola people | 69 |
| Aeta people | 244 | Tuscarora people | 117 | Idoma people | 69 |
| Kankanaey people | 242 | Wolof people | 116 | Tharu people | 69 |
| Snohomish people | 240 | Uyghurs | 115 | Afrikaners | 68 |
| Yi people | 235 | Mizo people | 114 | Jingpo people | 68 |
| Hazaras | 232 | Zulu people | 112 | Kipsigis people | 68 |
| Yaruro people | 228 | Fon people | 109 | Luo people | 67 |
| Temuan people | 226 | Maya peoples | 108 | Naso people | 67 |
| Ho people | 222 | Toubou people | 108 | Culture of Mauritius | 66 |
| Squamish people | 221 | Khasi people | 105 | Sukuma people | 66 |
| Kongo people | 216 | Arvanites | 104 | Belarusians | 65 |
| Thracians | 215 | Konkani people | 104 | Belizean Creole people | 64 |
| Wyandot people | 214 | Hmong people | 100 | Hausa people | 64 |
| Chams | 212 | Ewe people | 98 | Scottish people | 63 |
| Cappadocian Greeks | 208 | Punjabis | 98 | Mende people | 62 |
| Hadza people | 201 | Irish Travellers | 98 | Swedish-speaking Finns | 61 |
| Tumbuka people | 196 | Afghanistan Ethnic groups | 96 | Nuer people | 61 |
| Mon people | 189 | Toba Batak people | 96 | Tzeltal people | 61 |
| Karbi people | 184 | Somali people | 96 | Cornish people | 60 |
| Gondi people | 184 | Ket people | 94 | Wik-Mungkan people | 60 |
| Zaramo people | 179 | Melanau people | 93 | Iranian Azerbaijanis | 59 |
| Pame people | 177 | Bari people | 92 | Mentawai people | 59 |
| Mro-Khimi people | 172 | Bai people | 90 | Paiwan people | 58 |
| Pacific Northwest Coast Indigenes | 165 | Chuvash people | 90 | Turkish people | 58 |
| Jakun people | 165 | Omagua people | 90 | Gedeo people | 57 |
| Akan people | 164 | Istro-Romanians | 90 | Atayal people | 57 |
| Pojulu people | 164 | Baga people | 89 | Duala people | 56 |
| Kashubians | 163 | Darlong people | 89 | Mongolic peoples | 56 |
| Polynesians | 162 | Sierra Leone Creole people | 89 | Nivkh people | 56 |
| Urapmin people | 161 | Tamils | 89 | Turkmens | 55 |
| Kamba people | 160 | Zhuang people | 89 | Kiga people | 53 |
| Oromo people | 156 | Azerbaijanis | 88 | Dusun people | 53 |
| Senufo people | 156 | Harari people | 87 | Lao people | 53 |
| Louisiana Creole people | 154 | Turkana people | 87 | Chamorro people | 52 |
| Egyptians | 153 | Xhosa people | 85 | Ovambo people | 51 |
| Mandinka people | 153 | Assyrian people | 83 | Volga Tatars | 51 |
| Ga-Adangbe people | 152 | Bariba people | 82 | Ingrian Finns | 50 |
| Damara people | 152 | Gagauz people | 82 | Kikuyu people | 50 |
| Igbo people | 142 | Samoans | 82 | Karen people | 50 |
| Asmat people | 141 | Sylhetis | 82 | Khmer people | 48 |
| List of Igbo people | 141 | Tat people (Caucasus) | 78 | Romani people | 48 |

Table 13: The *# of documents* and *cultural knowledge assertion sentences* per **culture by ethnolinguistic group**. Please note that due to spacing, the top 171 ethnolingusitic cultural groups are included in this table only, but the full expanded version of this table is included in our code repository, which is also attached in the submission.

