# OpenReview forum: "NormLens: Massively Multicultural MLLM Reasoning with Fine-Grained Social Awareness"
_colmweb.org/COLM/2025/Workshop/Social_Sim — Social Sim'25_

### Official Review · Reviewer_2egJ · 2025-07-17

**Rating:** 4
**Overall Assessment:** 2
**Confidence:** 4

**Review:**

This submission unfortunately suffers from a critical formatting issue: it does not adhere to the specified COLM conference template, instead appearing to use the ACL template. Adherence to conference-specific formatting guidelines is a fundamental requirement for submission, as it ensures consistency, readability, and fair evaluation across all papers.

Due to this non-compliance with the submission guidelines, a detailed technical review of the paper's content, including its specific strengths, weaknesses, originality, and significance, cannot be adequately performed or is withheld at this stage. It is imperative that the authors reformat their submission strictly according to the COLM conference template before a proper review can proceed.

**Comments Suggestions And Typos:**

N/A

**Paper Summary:**

This paper introduces NormLens, a novel dataset and benchmark designed to assess and improve the fine-grained social awareness and multicultural reasoning capabilities of Multimodal Large Language Models (MLLMs). The core problem addressed is the prevalent cultural bias and lack of nuanced cultural commonsense knowledge in existing MLLMs, particularly regarding fine-grained situational contexts across diverse global cultures. The authors propose a new approach for massively multicultural MLLM knowledge acquisition at a fine-grained social awareness level.

**Relevance:**

4

**Summary Of Strengths:**

N/A

**Summary Of Weaknesses:**

N/A

---

### Official Review · Reviewer_BMoU · 2025-07-18

**Rating:** 7
**Overall Assessment:** 4
**Confidence:** 4

**Review:**

The paper presents a clear contribution and introduces a valuable resource and framework. There is rigorous human validation and well-designed experiments.

**Comments Suggestions And Typos:**

see Weaknesses

**Ethical Concerns:**

Data and code is not released.

**Paper Summary:**

MLLMs often lack fine-grained cultural commonsense knowledge and exhibit cultural biases. The paper introduces "NormLens" as an approach for massively multicultural MLLM knowledge acquisition. Its core contributions are (1) the normlens dataset with 42,000 assertions covering a large group of regions, (2) The MM-ACE framework for multimodal cultural knowledge acquisition, finetuning on contrastive (norm, dialogue, image) triplets. Experiments show MM-ACE improves cultural norm violation detection by 7.5% F-score. The study also reveals performance variations across cultures and topics, noting that existing human feedback alignment methods don't consistently improve fine-grained multicultural reasoning.

**Relevance:**

4

**Summary Of Strengths:**

1. Introduces NormLens, a large-scale, richly annotated dataset.
2. Thorough methods for data collection, including LLM prompting for profile extraction and robust negative sample generation.
3. MM-ACE demonstrates measurable improvements in cultural norm violation detection.
4. Provides valuable insights into MLLM performance variations across different cultures and topics.

**Summary Of Weaknesses:**

1. The paper reports a 7.5% F-score improvement for MM-ACE; a more granular error analysis on the types of cultural norm violations that MM-ACE still struggles with would offer deeper insights into its limitations and guide future improvements.
2. The paper identifies that LLM performance significantly decreases when probing finer-grained cultural information. While this highlights the challenge, the paper could benefit from further discussion or outlining potential strategies within the MM-ACE framework or future work to specifically address and mitigate this performance drop.

---

### Meta-Review · Program_Chairs · 2025-07-24

**Recommendation:** Accept

**Metareview:**

--